# Deep Partition Aggregation: Provable Defenses against General Poisoning Attacks

**Alexander Levine & Soheil Feizi**
Department of Computer Science
University of Maryland
College Park, MD 20742, USA
{alevine0, sfeizi}@cs.umd.edu

## Abstract

Adversarial poisoning attacks distort training data in order to corrupt the test-time behavior of a classifier. A provable defense provides a *certificate* for each test sample, which is a lower bound on the magnitude of any adversarial distortion of the training set that can corrupt the test sample's classification. We propose two novel provable defenses against poisoning attacks: (i) Deep Partition Aggregation (DPA), a certified defense against a *general* poisoning threat model, defined as the insertion or deletion of a bounded number of samples to the training set — by implication, this threat model also includes arbitrary distortions to a bounded number of images and/or labels; and (ii) Semi-Supervised DPA (SS-DPA), a certified defense against label-flipping poisoning attacks. DPA is an ensemble method where base models are trained on partitions of the training set determined by a hash function. DPA is related to both *subset aggregation*, a well-studied ensemble method in classical machine learning, as well as to *randomized smoothing*, a popular provable defense against evasion (inference) attacks. Our defense against label-flipping poison attacks, SS-DPA, uses a semi-supervised learning algorithm as its base classifier model: each base classifier is trained using the entire unlabeled training set in addition to the labels for a partition. SS-DPA significantly outperforms the existing certified defense for label-flipping attacks (Rosenfeld et al., 2020) on both MNIST and CIFAR-10: provably tolerating, for at least half of test images, over 600 label flips (vs. $< 200$ label flips) on MNIST and over 300 label flips (vs. 175 label flips) on CIFAR-10. Against general poisoning attacks where *no* prior certified defenses exists, DPA can certify $\geq 50\%$ of test images against over 500 poison image insertions on MNIST, and nine insertions on CIFAR-10. These results establish new state-of-the-art provable defenses against general and label-flipping poison attacks. Code is available at https://github.com/alevine0/DPA.

## 1 Introduction

Adversarial poisoning attacks are an important vulnerability in machine learning systems. In these attacks, an adversary can manipulate the training data of a classifier, in order to change the classifications of specific inputs at test time. Several poisoning threat models have been studied in the literature, including threat models where the adversary may insert new poison samples (Chen et al., 2017), manipulate the training labels (Xiao et al., 2012; Rosenfeld et al., 2020), or manipulate the training sample values (Biggio et al., 2012; Shafahi et al., 2018). A certified defense against a poisoning attack provides a *certificate* for each test sample, which is a guaranteed lower bound on the magnitude of any adversarial distortion of the training set that can corrupt the test sample's classification. In this work, we propose certified defenses against two types of poisoning attacks:

**General poisoning attacks:** In this threat model, the attacker can insert or remove a bounded number of samples from the training set. In particular, the attack magnitude $\rho$ is defined as the cardinality of the *symmetric difference* between the clean and poisoned training sets. This threat

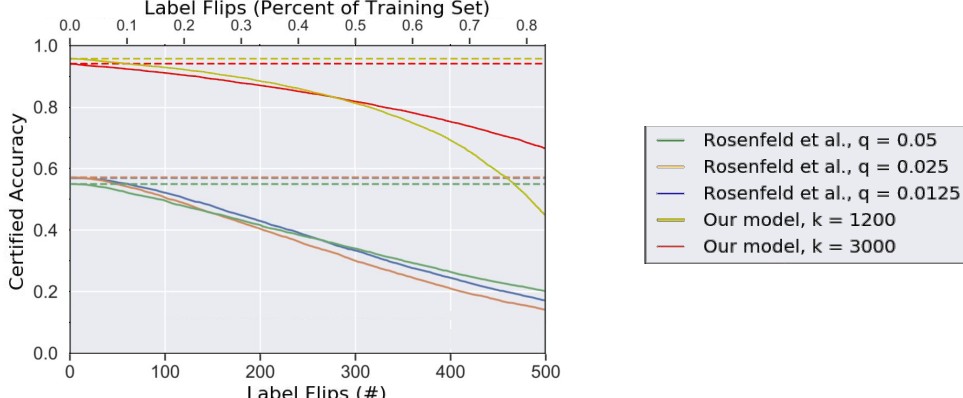

Figure 1: Comparison of certified accuracy to label-flipping poison attacks for our defense (SS-DPA algorithm) vs. Rosenfeld et al. (2020) on MNIST. Solid lines represent certified accuracy as a function of attack size; dashed lines show the clean accuracies of each model. Our algorithm produces substantially higher certified accuracies. Curves for Rosenfeld et al. (2020) are adapted from Figure 1 in that work. The parameter $q$ is a hyperparameter of Rosenfeld et al. (2020)'s algorithm, and $k$ is a hyperparameter of our algorithm: the number of base classifiers in an ensemble.

model also includes any distortion to an sample and/or label in the training set — a distortion of a training sample is simply the removal of the original sample followed by the insertion of the distorted sample. (Note that a sample distortion or label flip therefore increases the symmetric difference attack magnitude by two.)

**Label-flipping poisoning attacks:** In this threat model, the adversary changes only the label for $\rho$ out of $m$ training samples. Rosenfeld et al. (2020) has recently provided a certified defense for this threat model, which we improve upon.

In the last couple of years, certified defenses have been extensively studied for evasion attacks, where the adversary manipulates the test samples, rather than the training data (e.g. Wong & Kolter (2018); Gowal et al. (2018); Lecuyer et al. (2019); Li et al. (2018); Salman et al. (2019); Levine & Feizi (2020a;b); Cohen et al. (2019), etc.) In the evasion case, a certificate is a lower bound on the distance from the sample to the classifier's decision boundary: this guarantees that the sample's classification remains unchanged under adversarial distortions up to the certified magnitude.

Rosenfeld et al. (2020) provides an analogous certificate for label-flipping poisoning attacks: for an input sample $x$, the certificate of $x$ is a lower bound on the number of labels in the training set that would have to change in order to change the classification of $x$.[1] Rosenfeld et al. (2020)'s method is an adaptation of a certified defense for sparse ($L_0$) evasion attacks proposed by Lee et al. (2019). The adapted method for label-flipping attacks proposed by Rosenfeld et al. (2020) is equivalent to randomly flipping each training label with fixed probability and taking a consensus result. If implemented directly, this would require one to train a large ensemble of classifiers on different noisy versions of the training data. However, instead of actually doing this, Rosenfeld et al. (2020) focuses only on linear classifiers and is therefore able to analytically calculate the expected result. This gives deterministic, rather than probabilistic, certificates. Further, because Rosenfeld et al. (2020) considers a threat model where only labels are modified, they are able to train an *unsupervised* nonlinear feature extractor on the (unlabeled) training data before applying their technique, in order to learn more complex features.

Inspired by an improved provable defense against $L_0$ evasion attacks (Levine & Feizi, 2020a), in this paper, we develop certifiable defenses against general and label-flipping poisoning attacks that significantly outperform the current state-of-the-art certifiable defenses. In particular, we develop

---

[1]Steinhardt et al. (2017) also refers to a "certified defense" for poisoning attacks. However, the definition of the certificate is substantially different in that work, which instead provides overall accuracy guarantees under the assumption that the training and test data are drawn from similar distributions, rather than providing guarantees for individual realized inputs.

a certifiable defense against *general* poisoning attacks called **Deep Partition Aggregation (DPA)** which is based on partitioning the training set into $k$ partitions, with the partition assignment for a training sample determined by a hash function of the sample. The hash function can be any deterministic function that maps a training sample $\mathbf{t}$ to a partition assignment: the only requirement is that the hash value depends only on the value of the training sample $\mathbf{t}$ itself, so that neither poisoning other samples, nor changing the total number of samples, nor reordering the samples can change the partition that $\mathbf{t}$ is assigned to. We then train $k$ base classifiers separately, one on each partition. At the test time, we evaluate each of the base classifiers on the test sample $x$ and return the plurality classification $c$ as the final result. The key insight is that removing a training sample, or adding a new sample, will only change the contents of one partition, and therefore will only affect the classification of one of the $k$ base classifiers. This immediately leads to robustness certifications against general poisoning attacks which, to the best of our knowledge, is the first one of this kind.

If the adversary is restricted to flipping labels only (as in Rosenfeld et al. (2020)), we can achieve even larger certificates through a modified technique. In this setting, the unlabeled data is trustworthy: each base classifier in the ensemble can then make use of the entire training set without labels, but only has access to the labels in its own partition. Thus, each base classifier can be trained as if the entire dataset is available as unlabeled data, but only a very small number of labels are available. This is precisely the problem statement of semi-supervised learning (Verma et al., 2019; Luo et al., 2018; Laine & Aila, 2017; Kingma et al., 2014; Gidaris et al., 2018). We can then leverage these existing semi-supervised learning techniques directly to improve the accuracies of the base classifiers in DPA. Furthermore, we can ensure that a particular (unlabeled) sample is assigned to the same partition regardless of label, so that only one partition is affected by a label flip (rather than possibly two). The resulting algorithm, **Semi-Supervised Deep Partition Aggregation (SS-DPA)** yields substantially increased certified accuracy against label-flipping attacks, compared to DPA alone and compared to the current state-of-the-art. Furthermore, while our method is de-randomized (as Rosenfeld et al. (2020) is) and therefore yields deterministic certificates, our technique does not require that the classification model be linear, allowing deep networks to be used.

On MNIST, SS-DPA substantially outperforms the existing state of the art (Rosenfeld et al., 2020) in defending against label-flip attacks: we certify at least half of images in the test set against attacks to over 600 (1.0%) of the labels in the training set, while still maintaining over 93% accuracy (See Figure 1, and Table 1). In comparison, Rosenfeld et al. (2020)'s method achieves less than 60% clean accuracy on MNIST, and most test images cannot be certified with the correct class against attacks of even 200 label flips. We are also the first work to our knowledge to certify against general poisoning attacks, including insertions and deletions of new training images: in this domain, we can certify at least half of test images against attacks consisting of over 500 arbitrary training image insertions or deletions. On CIFAR-10, a substantially more difficult classification task, we can certify at least half of test images against label-flipping attacks on over 300 labels using SS-DPA (versus 175 label-flips for (Rosenfeld et al., 2020)), and can certify at least half of test images against general poisoning attacks of up to nine insertions or deletions using DPA. To see how our method performs on datasets with larger numbers of classes, we also tested our methods on the German Traffic Sign Recognition Benchmark (Stallkamp et al., 2012), a task with 43 classes and on average $\approx 1000$ samples per class. Here, we are able to certify at least half of test images as robust to 176 label flips, or 20 general poisoning attacks. **These results establish new state-of-the-art in provable defenses against label-flipping and general poisoning attacks.**

## 2 Related Works

Levine & Feizi (2020a) propose a *randomized ablation* technique to certifiably defend against sparse atatcks. Their method *ablates* some pixels, replacing them with a null value. Since it is possible for the base classifier to distinguish exactly which pixels originate from $x$, this results in more accurate base classifications and therefore substantially greater certified robustness than Lee et al. (2019). For example, on ImageNet, Lee et al. (2019) certifies the median test image against distortions of one pixel, while Levine & Feizi (2020a) certifies against distortions of 16 pixels.

Our proposed method is related to classical ensemble approaches in machine learning, namely bootstrap aggregation and subset aggregation (Breiman, 1996; Buja & Stuetzle, 2006; Bühlmann, 2003; Zaman & Hirose, 2009). However, in these methods each base classifier in the ensemble is trained

on an independently sampled collection of points from the training set: multiple classifiers in the ensemble may be trained on (and therefore poisoned by) the same sample point. The purpose of these methods has typically been to improve generalization. Bootstrap aggregation has been proposed as an *empirical* defense against poisoning attacks (Biggio et al., 2011) as well as for evasion attacks (Smutz & Stavrou, 2016). However, at the time of the initial distribution of this work, these techniques had not yet been used to provide *certified* robustness.[2] Our unique *partition aggregation* variant provides deterministic robustness certificates against poisoning attacks. See Appendix D for further discussion.

Weber et al. (2020) have recently proposed a different randomized-smoothing based defense against poisoning attacks by directly applying Cohen et al. (2019)'s smoothing $L_2$ evasion defense to the poisoning domain. The proposed technique can only certify for clean-label attacks (where only the existing samples in the dataset are modified, and not their labels), and the certificate guarantees robustness only to bounded $L_2$ distortions of the training data, where the $L_2$ norm of the distortion is calculated across all pixels in the *entire* training set. Due to well-known limitations of dimensional scaling for smoothing-based robustness certificates (Yang et al., 2020; Kumar et al., 2020; Blum et al., 2020), this yields certificates to only very small distortions of the training data. (For binary MNIST [13,007 images], the maximum reported $L_2$ certificate is 2 pixels.) Additionally, when using deep classifiers, Weber et al. (2020) proposes a randomized certificate, rather than a deterministic one, with a failure probability that decreases to zero only as the number of trained classifiers in an ensemble approaches infinity. Moreover, in Weber et al. (2020), unlike in our method, each classifier in the ensemble must be trained on a noisy version of the entire dataset. These issues hinder Weber et al. (2020)'s method to be an effective scheme for certified robustness against poisoning attacks. After the initial distribution of this work, a recent revision of Rosenfeld et al. (2020) has suggested using randomised smoothing techniques on training samples, rather than just training labels, as a general approach to poisoning defense. Both this work and Weber et al. (2020) could be considered as implementations of this idea, although this generalized proposal in Rosenfeld et al. (2020) does not include a derandomization scheme (unlike Rosenfeld et al. (2020)'s proposed derandomized defense against label-flipping attacks) .

Other prior works have provided *distributional*, rather than *pointwise* guarantees against poisoning attacks. In these works, there is a (high-probability) guarantee that the classifier will achieve a certain level of average overall accuracy on test data, under the assumption that the test data and clean (pre-poisoning) training data are drawn from the same distribution. These works do not provide any guarantees that apply to specific test samples, however. As mentioned above, Steinhardt et al. (2017) provides such a distributional guarantee, specifically for a threat model of addition of poison samples. Other such works include Sloan (1995), which considers label-flipping attacks and determines conditions under which PAC-learning is possible in the presence of such attacks, and Bshouty et al. (2002) provides similar guarantees for replacement of samples. Other works (Diakonikolas et al., 2016; Lai et al., 2016) provide distributional guarantees for unsupervised learning under poisoning attacks. Mahloujifar et al. (2019) proposes provably effective poisoning *attacks* with high-probability pointwise guarantees of effectiveness on test samples. However, that work relies on properties of the distribution that the training set is drawn from. Diakonikolas et al. (2019) provide a robust training algorithm which provably approximates the clean trained model despite poisoning (rather than the behavior at a certain test point): this result also makes assumptions about the distribution of the clean training data.

## 3 PROPOSED METHODS

### 3.1 NOTATION

Let $\mathcal{S}$ be the space of all possible unlabeled samples (i.e., the set of all possible images). We assume that it is possible to sort elements of $\mathcal{S}$ in a deterministic, unambiguous way. In particular, in the case of image data, we can sort images lexicographically by pixel values: in general, any data that can be represented digitally will be sortable. We represent labels as integers, so that the set of all possible labeled samples is $\mathcal{S}_L := \{(\mathbf{x}, c) | \mathbf{x} \in \mathcal{S}, c \in \mathbb{N}\}$. A training set for a classifier is then represented

---

[2]In a concurrent work, Jia et al. (2020) consider using bootstrap aggregation directly for certified robustness. Their certificates are therefore probabilistic, and are not as large, in median, as the certificates reported here for MNIST and CIFAR-10.

as $T \in \mathcal{P}(\mathcal{S}_L)$, where $\mathcal{P}(\mathcal{S}_L)$ is the power set of $\mathcal{S}_L$. For $\boldsymbol{t} \in \mathcal{S}_L$, we let **sample**$(\boldsymbol{t}) \in \mathcal{S}$ refer to the (unlabeled) sample, and **label**$(\boldsymbol{t}) \in \mathbb{N}$ refer to the label. For a set of samples $T \in \mathcal{P}(\mathcal{S}_L)$, we let **samples**$(T) \in \mathcal{P}(\mathcal{S})$ refer to the set of unique unlabeled samples which occur in $T$. A classifier model is defined as a deterministic function from *both* the training set *and* the sample to be classified to a label, i.e. $f : \mathcal{P}(\mathcal{S}_L) \times \mathcal{S} \to \mathbb{N}$. We will use $f(\cdot)$ to represent a base classifier model (i.e., a neural network), and $g(\cdot)$ to refer to a robust classifier (using DPA or SS-DPA).

$A \ominus B$ represents the set symmetric difference between $A$ and $B$: $A \ominus B = (A \setminus B) \cup (B \setminus A)$. The number of elements in $A$ is $|A|$, $[n]$ is the set of integers 1 through $n$, and $\lfloor z \rfloor$ is the largest integer less than or equal to $z$. $\mathbb{1}$ represents the indicator function: $\mathbb{1}_{\text{Prop}} = 1$ if Prop is true; $\mathbb{1}_{\text{Prop}} = 0$ otherwise. For a set $A$ of sortable elements, we define **Sort**$(A)$ as the sorted list of elements. For a list $L$ of unique elements, for $\boldsymbol{l} \in L$, we will define **index**$(L, l)$ as the index of $l$ in the list $L$.

## 3.2 DPA

The Deep Partition Aggregation (DPA) algorithm requires a base classifier model $f : \mathcal{P}(\mathcal{S}_L) \times \mathcal{S} \to \mathbb{N}$, a training set $T \in \mathcal{P}(\mathcal{S}_L)$, a deterministic hash function $h : \mathcal{S}_L \to \mathbb{N}$, and a hyperparameter $k \in \mathbb{N}$ indicating the number of base classifiers which will be used in the ensemble.

At the training time, the algorithm first uses the hash function $h$ to define partitions $P_1, ..., P_k \subseteq T$ of the training set, as follows:

$$P_i := \{\boldsymbol{t} \in T \,|\, h(\boldsymbol{t}) \equiv i \pmod{k}\}. \tag{1}$$

The hash function $h$ can be any deterministic function from $\mathcal{S}_L$ to $\mathbb{N}$: however, it is preferable that the partitions are roughly equal in size. Therefore we should choose an $h$ which maps samples to a domain of integers significantly larger than $k$, in a way such that $h(.) \pmod{k}$ will be roughly uniform over $[k]$. In practice, for image data, we let $h(\boldsymbol{t})$ be the sum of the pixel values in the image $\boldsymbol{t}$.

Base classifiers are then trained on each partition: we define trained base classifiers $f_i : \mathcal{S} \to \mathbb{N}$ as:

$$f_i(\boldsymbol{x}) := f(P_i, \boldsymbol{x}). \tag{2}$$

Finally, at the inference time, we evaluate the input on each base classification, and then count the number of classifiers which return each class:

$$n_c(\boldsymbol{x}) := |\{i \in [k] \,|\, f_i(\boldsymbol{x}) = c\}|. \tag{3}$$

This lets us define the classifier which returns the consensus output of the ensemble:

$$g_{\textbf{dpa}}(T, \boldsymbol{x}) := \arg\max_c n_c(\boldsymbol{x}). \tag{4}$$

When taking the argmax, we break ties deterministically by returning the smaller class index. The resulting robust classifier has the following guarantee:

**Theorem 1.** *For a fixed deterministic base classifier $f$, hash function $h$, ensemble size $k$, training set $T$, and input $\boldsymbol{x}$, let:*

$$c := g_{\textit{dpa}}(T, \boldsymbol{x})$$
$$\bar{\rho}(\boldsymbol{x}) := \left\lfloor \frac{n_c - \max_{c' \neq c}(n_{c'}(\boldsymbol{x}) + \mathbb{1}_{c' < c})}{2} \right\rfloor. \tag{5}$$

*Then, for any poisoned training set $U$, if $|T \ominus U| \leq \bar{\rho}(\boldsymbol{x})$, then $g_{\textit{dpa}}(U, \boldsymbol{x}) = c$.*

All proofs are presented in Appendix A. Note that $T$ and $U$ are *unordered* sets: therefore, in addition to providing certified robustness against insertions or deletions of training data, the robust classifier $g_{\textbf{dpa}}$ is also invariant under re-ordering of the training data, provided that $f$ has this invariance (which is implied, because $f$ maps deterministically from a set; see Section 3.2.1 for practical considerations). As mentioned in Section 1, DPA is a deterministic variant of randomized ablation (Levine & Feizi, 2020a) adapted to the poisoning domain. Each base classifier ablates most of the training set, retaining only the samples in one partition. However, unlike in randomized ablation, the partitions are deterministic and use disjoint samples, rather than selecting them randomly and independently. In Appendix C, we argue that our derandomization has little effect on the certified accuracies, while allowing for exact certificates using finite samples. We also discuss how this work relates to Levine & Feizi (2020c), which proposes a de-randomized ablation technique for a restricted class of sparse evasion attacks (patch adversarial attacks).

### 3.2.1 DPA PRACTICAL IMPLEMENTATION DETAILS

One of the advantages of DPA is that we can use deep neural networks for the base classifier $f$. However, enforcing that the output of a deep neural network is a deterministic function of its training data, and specifically, its training data as an unordered set, requires some care. First, we must remove dependence on the order in which the training samples are read in. To do this, in each partition $P_i$, we sort the training samples prior to training, taking advantage of the assumption that $\mathcal{S}$ is well-ordered (and therefore $\mathcal{S}_L = \mathcal{S} \times \mathbb{N}$ is also well ordered). In the case of the image data, this is implemented as a lexical sort by pixel values, with the labels concatenated to the samples as an additional value. The training procedure for the network, which is based on standard stochastic gradient descent, must also be made deterministic: in our PyTorch (Paszke et al., 2019) implementation, this can be accomplished by deterministically setting a random seed at the start of training. As discussed in Appendix F, we find that it is best for the final classifier accuracy to use different random seeds during training for each partition. This reduces the correlation in output between base classifiers in the ensemble. Thus, in practice, we use the partition index as the random seed (i.e., we train base classifier $f_i$ using random seed $i$.)

### 3.3 SS-DPA

Semi-Supervised DPA (SS-DPA) is a defense against label-flip attacks. In SS-DPA, the base classifier may be a *semi-supervised* learning algorithm: it can use the entire unlabeled training dataset, in addition to the labels for a partition. We will therefore define the base classifier to also accept an unlabelled dataset as input: $f : \mathcal{P}(\mathcal{S}) \times \mathcal{P}(\mathcal{S}_L) \times \mathcal{S} \to \mathbb{N}$. Additionally, our method of partitioning the data is modified both to ensure that changing the label of a sample affects only one partition rather than possibly two, and to create a more equal distribution of samples between partitions.

First, we will sort the unlabeled data **samples**$(T)$:

$$T_{\text{sorted}} := \textbf{Sort}(\textbf{samples}(T)). \tag{6}$$

For a sample $\boldsymbol{t} \in T$, note that $\textbf{index}(T_{\text{sorted}}, \textbf{sample}(\boldsymbol{t}))$ is invariant under any label-flipping attack to $T$, and also under permutation of the training data as they are read. We now partition the data based on sorted index:

$$P_i := \{\boldsymbol{t} \in T \,|\, \textbf{index}(T_{\text{sorted}}, \textbf{sample}(\boldsymbol{t})) \equiv i \pmod{k}\}. \tag{7}$$

Note that in this partitioning scheme, we no longer need to use a hash function $h$. Moreover, this scheme creates a more uniform distribution of samples between partitions, compared with the hashing scheme used in DPA. This can lead to improved certificates: see Appendix E. This sorting-based partitioning is possible because the unlabeled samples are "clean", so we can rely on their ordering, when sorted, to remain fixed. As in DPA, we train base classifiers on each partition, this time additionally using the entire unlabeled training set:

$$f_i(\boldsymbol{x}) := f(\textbf{samples}(T), P_i, \boldsymbol{x}). \tag{8}$$

The inference procedure is the same as in the standard DPA:

$$n_c(\boldsymbol{x}) := |\{i \in [k] \,|\, f_i(\boldsymbol{x}) = c\}|$$
$$g_{\textbf{ssdpa}}(T, \boldsymbol{x}) := \arg\max_c n_c(\boldsymbol{x}) \tag{9}$$

The SS-DPA algorithm provides the following robustness guarantee against label-flipping attacks.[3]

**Theorem 2.** *For a fixed deterministic semi-supervised base classifier $f$, ensemble size $k$, training set $T$ (with no repeated samples), and input $\boldsymbol{x}$, let:*

$$c := g_{\textit{ssdpa}}(T, \boldsymbol{x}),$$
$$\bar{\rho}(\boldsymbol{x}) := \left\lfloor \frac{n_c - \max_{c' \neq c}(n_{c'}(\boldsymbol{x}) + \mathbb{1}_{c' < c})}{2} \right\rfloor. \tag{10}$$

*For a poisoned training set $U$ obtained by changing the labels of at most $\bar{\rho}$ samples in $T$, $g_{\textit{ssdpa}}(U, \boldsymbol{x}) = c$.*

---

[3]The theorem as stated assumes that there are no repeated unlabeled samples (with different labels) in the training set $T$. This is a reasonable assumption, and in the label-flipping attack model, the attacker cannot cause this assumption to be broken. Without this assumption, the analysis is more complicated; see Appendix G.

| | Training set size | Number of Partitions $k$ | Median Certified Robustness | Clean Accuracy | Base Classifier Accuracy | Training time per Partition |
|---|---|---|---|---|---|---|
| **MNIST, DPA** | 60000 | 1200 | 448 | 95.85% | 76.97% | 0.33 min |
| | | 3000 | 509 | 93.36% | 49.54% | 0.27 min |
| **MNIST, SS-DPA** | 60000 | 1200 | 485 | 95.62% | 80.77% | 0.15 min |
| | | 3000 | 645 | 93.90% | 57.65% | 0.16 min |
| **CIFAR, DPA** | 50000 | 50 | 9 | 70.16% | 56.39% | 1.49 min |
| | | 250 | 5 | 55.65% | 35.17% | 0.58 min |
| | | 1000 | N/A | 44.52% | 23.20% | 0.30 min |
| **CIFAR, SS-DPA** | 50000 | 50 | 25 | 90.89% | 89.06% | 0.94 min |
| | | 250 | 124 | 90.33% | 86.25% | 0.43 min |
| | | 1000 | 392 | 89.02% | 75.83% | 0.33 min |
| **GTSRB, DPA** | 39209 | 50 | 20 | 89.20% | 73.94% | 2.64 min |
| | | 100 | 4 | 55.90% | 35.64% | 1.60 min |
| **GTSRB, SS-DPA** | 39209 | 50 | 25 | 97.09% | 96.35% | 2.73 min |
| | | 100 | 50 | 96.76% | 94.96% | 1.56 min |
| | | 200 | 99 | 96.34% | 91.54% | 1.23 min |
| | | 400 | 176 | 95.80% | 83.60% | 0.78 min |

Table 1: Summary statistics for DPA and SS-DPA algorithms on MNIST, CIFAR, and GTSRB. Median Certified Robustness is the attack magnitude (symmetric difference for DPA, label flips for SS-DPA) at which certified accuracy is 50%. Training times are on a single GPU; note that many partitions can be trained in parallel. Note we observe some constant overhead time for training each classifier, so on MNIST, where the training time per image is small, $k$ has little effect on the training time. For SS-DPA, training times do not include the time to train the unsupervised feature embedding (which must only be done once).

### 3.3.1 SEMI-SUPERVISED LEARNING METHODS FOR SS-DPA

In the standard DPA algorithm, we are able to train each classifier in the ensemble using only a small fraction of the training data; this means that each classifier can be trained relatively quickly: as the number of classifiers increases, the time to train each classifier can decrease (see Table 1). However, in a naive implementation of SS-DPA, Equation 8 might suggest that training time will scale with $k$, because each semi-supervised base classifier requires to be trained on the entire training set. Indeed, with many popular and highly effective choices of semi-supervised classification algorithms, such as temporal ensembling (Laine & Aila, 2017), ICT (Verma et al., 2019), Teacher Graphs (Luo et al., 2018) and generative approaches (Kingma et al., 2014), the main training loop trains on both labeled and unlabeled samples, so we would see the total training time scale linearly with $k$. In order to avoid this, we instead choose a semi-supervised training method where the unlabeled samples are used *only* to learn semantic features of the data, *before* the labeled samples are introduced: this allows us to use the unlabeled samples only once, and to then share the learned feature representations when training each base classifier. In our experiments, we choose RotNet (Gidaris et al., 2018) for experiments on MNIST, and SimCLR (Chen et al., 2020) for experiments on CIFAR-10 and GTSRB. Both methods learn an unsupervised embedding of the training set, on top of which all classifiers in the ensemble can be learned. Note that Rosenfeld et al. (2020) also uses SimCLR for CIFAR-10 experiments. As discussed in Section 3.2.1, we also sort the data prior to learning (including when learning unsupervised features), and set random seeds, in order to ensure determinism.

## 4 RESULTS

In this section, we present empirical results evaluating the performance of proposed methods, DPA and SS-DPA, against poison attacks on MNIST, CIFAR-10, and GTSRB datasets. As discussed in Section 3.3.1, we use the RotNet architecture (Gidaris et al., 2018) for SS-DPA's semi-supervised learning on MNIST. Conveniently, the RotNet architecture is structured such that the feature extracting layers, combined with the final classification layers, together make up the Network-In-Network (NiN) architecture for the supervised classification (Lin et al., 2013). Therefore, on MNIST, we

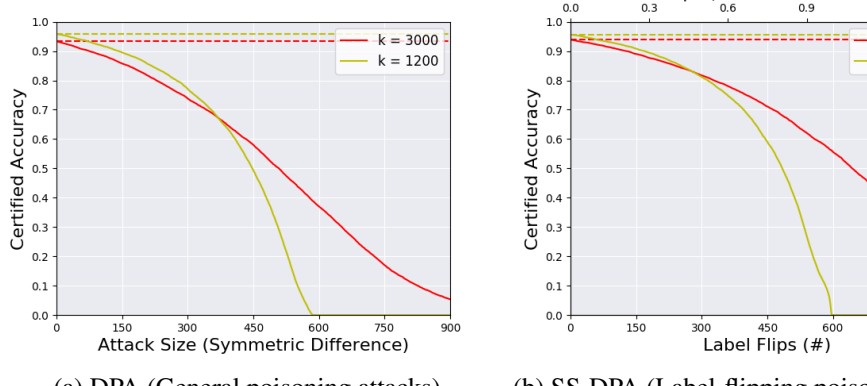

(a) DPA (General poisoning attacks)      (b) SS-DPA (Label-flipping poisoning attacks)

Figure 2: Certified Accuracy to poisoning attacks on MNIST, using (a) DPA to certify against general poisoning attacks, and (b) SS-DPA to certify against label-flipping attacks. Dashed lines show the clean accuracies of each model.

use NiN for DPA's supervised training, and RotNet for SS-DPA's semi-supervised training. We use training parameters, for both the DPA (NiN) and SS-DPA (RotNet), directly from Gidaris et al. (2018), with a slight modification: we eliminate horizontal flips in data augmentation, because horizontal alignment is semantically meaningful for digits.[4] On CIFAR-10 and GTSRB, we also use NiN (with full data augmentation for CIFAR-10, and without horizontal flips for GTSRB) for DPA experiments. For semi-supervised learning in SS-DPA, we use SimCLR (Chen et al., 2020) for both datasets, as (Rosenfeld et al., 2020) does on CIFAR-10. SimCLR hyperparameters are provided in Appendix I, and additional details about processing the GTSRB dataset are provided in Appendix J. Note that for SimCLR we use linear classifiers as the final, supervised classifiers for each partition.

Results are presented in Figures 2, 3, 4, and are summarized in Table 1. Our metric, Certified Accuracy as a function of attack magnitude (symmetric-difference or label-flips), refers to the fraction of samples which are both correctly classified and are certified as robust to attacks of that magnitude. Note that *different* poisoning perturbations, which poison different sets of training samples, may be required to poison *each* test sample; i.e. we assume the attacker can use the attack budget separately for each test sample. Table 1 also reports Median Certified Robustness, the attack magnitude to which at least 50% of the test set is provably robust.

Our SS-DPA method substantially outperforms the existing certificate (Rosenfeld et al., 2020) on label-flipping attacks: in median, 392 label flips on CIFAR-10, versus 175; 645 label flips on MNIST, versus < 200. With DPA, we are also able to certify at least half of MNIST images to attacks of over 500 poisoning insertions or deletions, and can certify at least half of CIFAR-10 images to 9 poisoning insertions or deletions. On GTSRB, we can certify over half of images to 20 poisoning insertions or deletions, or 176 label flips. Note that this represents a substantially larger fraction of each class (each class has < 1000 training images on average) compared to certificates on CIFAR-10 (5000 training images per class). See Appendix H for additional experiments on GTSRB.

The hyperparameter $k$ controls the number of classifiers in the ensemble: because each sample is used in training exactly one classifier, the average number of samples used to train each classifier is inversely proportional to $k$. Therefore, we observe that the base classifier accuracy (and therefore also the final ensemble classifier accuracy) decreases as $k$ is increased; see Table 1. However, because the certificates described in Theorems 1 and 2 depend directly on the gap in the *number* of classifiers in the ensemble which output the top and runner-up classes, larger numbers of classifiers are necessary to achieve large certificates. In fact, using $k$ classifiers, the largest certified robustness possible is $k/2$. Thus, we see in Figures 2, 3 and 4 that larger values of $k$ tend to produce larger robustness certificates. Therefore $k$ controls a trade-off between robustness and accuracy.

---

[4]In addition to the de-randomization changes mentioned in Section 3.2.1, we made one modification to the NiN 'baseline' for supervised learning: the baseline implementation in Gidaris et al. (2018), even when trained on a small subset of the training data, uses normalization constants derived from the entire training set. This is a (minor) error in Gidaris et al. (2018) that we correct by calculating normalization constants on each subset.

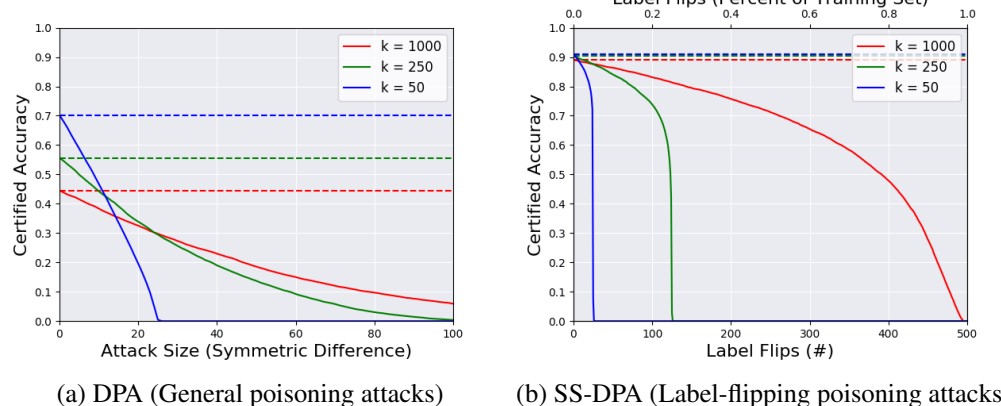

(a) DPA (General poisoning attacks)  (b) SS-DPA (Label-flipping poisoning attacks)

Figure 3: Certified Accuracy to poisoning attacks on CIFAR, using (a) DPA to certify against general poisoning attacks, and (b) SS-DPA to certify against label-flipping attacks.

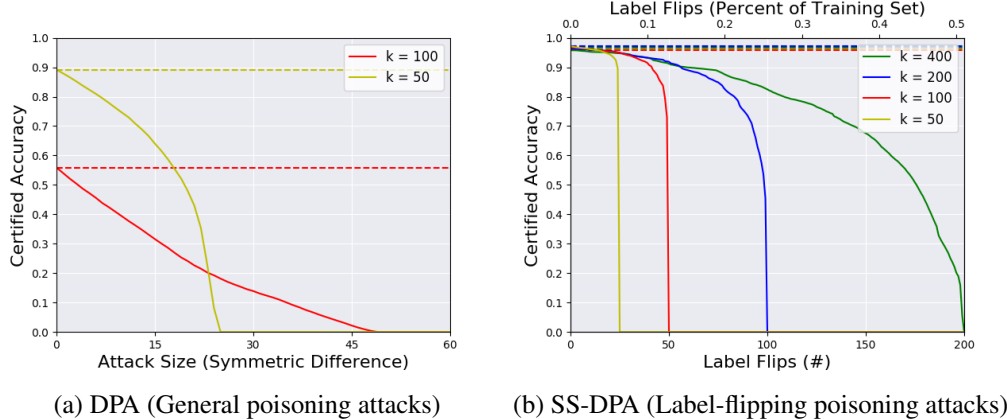

(a) DPA (General poisoning attacks)  (b) SS-DPA (Label-flipping poisoning attacks)

Figure 4: Certified Accuracy to poisoning attacks on GTSRB, using (a) DPA to certify against general poisoning attacks, and (b) SS-DPA to certify against label-flipping attacks.

Rosenfeld et al. (2020) also reports robustness certificates against label-flipping attacks on binary MNIST classification, with classes 1 and 7. Rosenfeld et al. (2020) reports clean-accuracy of 94.5% and certified accuracies for attack magnitudes up to 2000 label flips (out of 13007), with best certified accuracy less than 70%. By contrast, using a specialized form of SS-DPA, we are able to achieve clean accuracy of 95.5%, with every correctly-classified image certifiably robust up to 5952 label flips (i.e. certified accuracy is also 95.5% at 5952 label flips.) See Appendix B for discussion.

## 5 CONCLUSION

In this paper, we described a novel approach to provable defenses against poisoning attacks. Unlike previous techniques, our method both allows for exact, deterministic certificates and can be implemented using deep neural networks. These advantages allow us to outperform the current state-of-the-art on label-flip attacks, and to develop the first certified defense against a broadly defined class of *general* poisoning attacks.

## 6 ACKNOWLEDGEMENTS

This work was supported in part by NSF CAREER AWARD 1942230, HR 00111990077, HR 001119S0026, NIST 60NANB20D134 and Simons Fellowship on "Foundations of Deep Learning."

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

## A    PROOFS

**Theorem 1.** *For a fixed deterministic base classifier $f$, hash function $h$, ensemble size $k$, training set $T$, and input $x$, let:*

$$c := g_{dpa}(T, x)$$
$$\bar{\rho}(x) := \left\lfloor \frac{n_c - \max_{c' \neq c}(n_{c'}(x) + \mathbb{1}_{c' < c})}{2} \cdot \right\rfloor \qquad (11)$$

*Then, for any poisoned training set $U$, if $|T \ominus U| \leq \bar{\rho}(x)$, we have: $g_{dpa}(U, x) = c$.*

*Proof.* We define the partitions, trained classifiers, and counts for each training set ($T$ and $U$) as described in the main text:

$$P_i^T := \{t \in T \,|\, h(t) \equiv i \pmod{k}\}$$
$$P_i^U := \{t \in U \,|\, h(t) \equiv i \pmod{k}\} \qquad (12)$$

$$f_i^T(x) := f(P_i^T, x)$$
$$f_i^U(x) := f(P_i^U, x) \qquad (13)$$

$$n_c^T(x) := |\{i \in [k] \,|\, f_i^T(x) = c\}|$$
$$n_c^U(x) := |\{i \in [k] \,|\, f_i^U(x) = c\}| \qquad (14)$$

$$g_{\mathbf{dpa}}(T, x) := \arg\max_c n_c^T(x)$$
$$g_{\mathbf{dpa}}(U, x) := \arg\max_c n_c^U(x) \qquad (15)$$

Note that here, we are using superscripts to explicitly distinguish between partitions (as well as base classifiers and counts) of the clean training set $T$ and the poisoned dataset $U$ (i.e, $P_i^T$ is equivalent to $P_i$ in the main text). In Equation 15, as discussed in the main text, when taking the argmax, we break ties deterministically by returning the smaller class index.

Note that $P_i^T = P_i^U$ unless there is some $t$, with $h(t) \equiv i \pmod{k}$, in $T \ominus U$. Because the mapping from $t$ to $h(t) \pmod{k}$ is a deterministic function, the number of partitions $i$ for which $P_i^T \neq P_i^U$ is at most $|T \ominus U|$, which is at most $\bar{\rho}(x)$. $P_i^T = P_i^U$ implies $f_i^T(x) = f_i^U(x)$, so the number of classifiers $i$ for which $f_i^T(x) \neq f_i^U(x)$ is also at most $\bar{\rho}(x)$. Then:

$$\forall c' : |n_{c'}^T - n_{c'}^U| \leq \bar{\rho}(x). \qquad (16)$$

Let $c := g_{\mathbf{dpa}}(T, x)$. Note that $g_{\mathbf{dpa}}(U, x) = c$ iff:

$$\forall c' < c : n_c^U(x) > n_{c'}^U(x)$$
$$\forall c' > c : n_c^U(x) \geq n_{c'}^U(x)$$

where the separate cases come from the deterministic selection of the smaller index in cases of ties in Equation 15: this can be condensed to $\forall c' \neq c : n_c^U(x) \geq n_{c'}^U(x) + \mathbb{1}_{c' < c}$. Then by triangle inequality with Equation 16, we have that $g_{\mathbf{dpa}}(U, x) = c$ if $\forall c' \neq c : n_c^T(x) \geq n_{c'}^T(x) + 2\bar{\rho} + \mathbb{1}_{c' < c}$. This condition is true by the definition of $\bar{\rho}(x)$, so $g_{\mathbf{dpa}}(U, x) = c$. $\qquad \square$

**Theorem 2.** *For a fixed deterministic semi-supervised base classifier $f$, ensemble size $k$, training set $T$ (with no repeated samples), and input $\boldsymbol{x}$, let:*

$$c := g_{ssdpa}(T, \boldsymbol{x})$$

$$\bar{\rho}(\boldsymbol{x}) := \left\lfloor \frac{n_c - \max_{c' \neq c}(n_{c'}(\boldsymbol{x}) + \mathbb{1}_{c' < c})}{2} \right\rfloor \tag{17}$$

*For a poisoned training set $U$ obtained by changing the labels of at most $\bar{\rho}$ samples in $T$, $g_{ssdpa}(U, \boldsymbol{x}) = c$.*

*Proof.* Recall the definition:

$$T_{\text{sorted}} := \textbf{Sort}(\textbf{samples}(T)). \tag{18}$$

Because $\textbf{samples}(T) = \textbf{samples}(U)$, we have $T_{\text{sorted}} = U_{\text{sorted}}$. We can then define partitions and base classifiers for each training set ($T$ and $U$) as described in the main text:

$$P_i^T := \{\boldsymbol{t} \in T \mid \textbf{index}(T_{\text{sorted}}, \textbf{sample}(\boldsymbol{t})) \equiv i \pmod k\}$$
$$P_i^U := \{\boldsymbol{t} \in U \mid \textbf{index}(T_{\text{sorted}}, \textbf{sample}(\boldsymbol{t})) \equiv i \pmod k\} \tag{19}$$

$$f_i^T(\boldsymbol{x}) := f(\textbf{samples}(T), P_i^T, \boldsymbol{x})$$
$$f_i^U(\boldsymbol{x}) := f(\textbf{samples}(T), P_i^U, \boldsymbol{x}) \tag{20}$$

Recall that for any $\boldsymbol{t} \in T$, $\textbf{index}(T_{\text{sorted}}, \textbf{sample}(\boldsymbol{t}))$ is invariant under label-flipping attack to $T$. Then, for each $i$, the samples in $P_i^T$ will be the same as the samples in $P_i^U$, possibly with some labels flipped. In particular, the functions $f_i^T(\cdot)$ and $f_i^U(\cdot)$ will be identical, unless the label of some sample with $\textbf{index}(T_{\text{sorted}}, \textbf{sample}(\boldsymbol{t})) \equiv i \pmod k$ has been changed. If at most $\bar{\rho}(\boldsymbol{x})$ labels change, at most $\bar{\rho}(\boldsymbol{x})$ ensemble classifiers are affected: the rest of the proof proceeds similarly as that of Theorem 1. $\qquad\square$

## B  BINARY MNIST EXPERIMENTS

We perform a specialized instance of SS-DPA on the binary '1' versus '7' MNIST classification task. Specifically, we set $k = m$, so that every partition receives only one label. We first use 2-means clustering on the unlabeled data, to compute two means. This allows for each base classifier to use a very simple "semi-supervised learning algorithm": if the test image and the one labeled training image provided to the base classifier belong to the same cluster, then the base classifier assigns the label of the training image to the test image. Otherwise, it assigns the opposite label to the test image. Formally:

$$\mu_1, \mu_2 := \text{Cluster centroids of } \textbf{samples}(T)$$

$$\textbf{assignment}(\boldsymbol{s}) := \begin{cases} 1, & \text{if } \|\mu_1 - \boldsymbol{s}\|_2 \leq \|\mu_2 - \boldsymbol{s}\|_2 \\ 2, & \text{if } \|\mu_1 - \boldsymbol{s}\|_2 > \|\mu_2 - \boldsymbol{s}\|_2 \end{cases}$$

$$f_i(\boldsymbol{x}) = f(\textbf{samples}(T), \{\boldsymbol{t}_i\}, \boldsymbol{x}) = \begin{cases} \textbf{label}(\boldsymbol{t}_i), & \text{if } \textbf{assignment}(\boldsymbol{x}) = \textbf{assignment}(\boldsymbol{t}_i) \\ 1 - \textbf{label}(\boldsymbol{t}_i), & \text{if } \textbf{assignment}(\boldsymbol{x}) \neq \textbf{assignment}(\boldsymbol{t}_i) \end{cases}$$

Note that each base classifier behaves exactly identically, up to a transpose of the labels: so in practice, we simply count the training samples which associate each of the two cluster centroids with each of the two labels, and determine the number of label flips which would be required to change the consensus label assignments *of the clusters*. At the test time, each test image therefore needs to be processed only once. The amount of time required for inference is then simply the time needed to calculate the distance from the test sample to each of the two clusters. This also means that every image has the same robustness certificate. As stated in the main text, using this method, we are able to achieve clean accuracy of $95.5\%$, with every correctly-classified image certifiably robust up to 5952 label flips (i.e. certified accuracy is also $95.5\%$ at 5952 label flips.) This means that the classifier is robust to adversarial label flips on $45.8\%$ of the training data.

Rosenfeld et al. (2020) also reports robustness certificates against label-flipping attacks on binary MNIST classification with classes 1 and 7. Rosenfeld et al. (2020) reports clean-accuracy of $94.5\%$ and certified accuracies for attack magnitudes up to 2000 label flips (out of 13007: $15.4\%$), with the best certified accuracy less than $70\%$.

## C  RELATIONSHIP TO RANDOMIZED ABLATION

As mentioned in Section 1, (SS-)DPA is in some sense related to Randomized Ablation (Levine & Feizi, 2020a) (used in defense against sparse inference-time attacks) for training-time poisoning attacks. Randomized Ablation is a certified defense against $L_0$ (sparse) inference attacks, in which the final classification is a consensus among classifications of copies of the image. In each copy, a fixed number of pixels are randomly ablated (replaced with a null value). A direct application of Randomized Ablation to poisoning attacks would require each base classifier to be trained on a *random* subset of the training data, with each base classifier's training set chosen randomly and independently. Due to the randomized nature of this algorithm, estimation error would have to be considered in practice when applying Randomized Ablation using a finite number of base classifiers: this decreases the certificates that can be reported, while also introducing a failure probability to the certificates. By contrast, in our algorithms, the partitions are *deterministic* and use disjoint, rather than independent, samples. In this section, we argue that our derandomization has little effect on the certified accuracies compared to randomized ablation, even considering randomized ablation with no estimation error (i.e., with infinite base classifiers). In the poisoning case specifically, using additional base classifiers is expensive – because they must each be trained – so one would observe a large estimation error when using a realistic number of base classifiers. Therefore our derandomization can potentially improve the certificates which can be reported, while also allowing for exact certificates using a finite number of base classifiers.

For simplicity, consider the label-flipping case. In this case, the training set has a fixed size, $m$. Thus, Randomized Ablation bounds can be considered directly. A direct adaptation of Levine & Feizi (2020a) would, for each base classifier, choose $s$ out of $m$ samples to retain labels for, and would ablate the labels for the rest of the training data. Suppose an adversary has flipped $r$ labels. For each base classifier, the probability that a flipped label is used in classification (and therefore that the base classifier is 'poisoned') is:

$$\Pr(\text{poisoned})_{\text{RA}} = 1 - \frac{\binom{m-r}{s}}{\binom{m}{s}} \tag{21}$$

where "RA" stands for Randomized Ablation.

In this direct adaptation, one must then use a very large ensemble of randomized classifiers. The ensemble must be large enough that we can estimate with high confidence the probabilities (on the distribution of possible choices of training labels to retain) that the base classifier selects each class. If the gap between the highest and the next-highest class probabilities can be determined to be greater than $2\Pr(\text{poisoned})_{\text{RA}}$, then the consensus classification cannot be changed by flipping $r$ labels. This is because, at worst, every poisoned classifier could switch the highest-class classification to the runner-up class, reducing the gap by at most $2\Pr(\text{poisoned})$.

Note that this estimation relies on each base classifier using a subset of labels selected *randomly and independently* from the other base classifier. In contrast, our SS-DPA method selects each subset *disjointly*. If $r$ labels are flipped, assuming that in the worst case each flipped label is in a different partition, using the union bound, the proportion of base classifiers which can be poisoned is

$$\Pr(\text{poisoned})_{\text{SS-DPA}} \leq \frac{r}{k} = \frac{rs}{m} \tag{22}$$

where for simplicity we assume that $k$ evenly divides the number of samples $m$, so $s = m/k$ labels are kept by each partition. Again we need the gap in class probabilities to be at least $2\Pr(\text{poisoned})_{\text{SS-DPA}}$ to ensure robustness. While the use of the union bound might suggest that our deterministic scheme (Equation 22) might lead to a significantly looser bound than that of the probabilistic certificate (Equation 21), this is not the case in practice where $rs << m$. For example, in an MNIST-sized dataset ($m = 60000$), using $s = 50$ labels per base classifier, to certify for $r = 200$ label flips, we have $\Pr(\text{poisoned})_{\text{RA}} = 0.154$, and $\Pr(\text{poisoned})_{\text{SS-DPA}} \leq 0.167$. The derandomization only sightly increases the required gap between the top two class probabilities.

To understand this, note that if the number of poisonings $r$ is small compared to the number of partitions $k$, then even if the partitions are random and independent, the chance that any two poisonings occur in the same partition is quite small. In that case, the union bound in Equation 22 is actually quite close to an independence assumption. By accepting this small increase in the upper bound of

the probability that each base classification is poisoned, our method provides all of the benefits of de-randomization, including allowing for exact robustness certificates using only a finite number of classifiers. Additionally, note that in the Randomized Ablation case, the *empirical* gap in estimated class probabilities must be somewhat larger than $\Pr(\text{poisoned})_{\text{RA}}$ in order to certify robustness with high confidence, due to estimation error: the gap required increases more as the number of base classifiers decreases. This is particularly important in the poisoning case, because training a large number of classifiers is substantially more expensive than performing a large number of evaluations, as in randomized smoothing for evasion attacks.

We also note that Levine & Feizi (2020c) also used a de-randomized scheme based on Randomized Ablation to certifiably defend against evasion patch attacks. However, in that work, the de-randomization does not involve a union bound over *arbitrary* partitions of the vulnerable inputs. Instead, in the case of patch attacks, the attack is geometrically constrained: the image is therefore divided into geometric regions (bands or blocks) such that the attacker will only overlap with a fixed number of these regions. Each base classifier then uses only a *single* region to make its classification. Levine & Feizi (2020c) do not apply this to derandomization via disjoint subsets/union bound to defend against poison attacks. Also, we note that we borrow from Levine & Feizi (2020c) the deterministic "tie-breaking" technique when evaluating the consensus class in Equation 4, which can increase our robustness certificate by up to one.

## D    RELATIONSHIP TO EXISTING ENSEMBLE METHODS

As mentioned in Section 1, our proposed method is related to classical ensemble approaches in machine learning, namely bootstrap aggregation ("bagging") and subset aggregation ("subagging") (Breiman, 1996; Buja & Stuetzle, 2006; Bühlmann, 2003; Zaman & Hirose, 2009). In these methods, each base classifier in the ensemble is trained on an independently sampled collection of points from the training set: this means that multiple classifiers in the ensemble may be trained on the same sample point. The purpose of these methods has typically been to improve generalization, and therefore to improve test set accuracy: bagging and subagging decrease the variance component of the classifier's error.

In subagging, each training set for a base classifier is an independently sampled subset of the training data: this is in fact an identical formulation to the "direct Randomized Ablation" approach discussed in Appendix C. However, in practice, the size of each training subset has typically been quite large: the bias error term increases with decreasing subsample sizes (Buja & Stuetzle, 2006). Thus, the optimal subsample size for maximum accuracy is large: Bühlmann (2003) recommends using $s = m/2$ samples per classifier ("half-subagging"), with theoretical justification for optimal generalization. This would *not* be useful in Randomized Ablation-like certification, because any one poisoned element would affect half of the ensemble. Indeed, in our certifiably robust classifiers, we observe a trade-off between accuracy and certified robustness: our use of many very small partitions is clearly not optimal for the test-set accuracy (Table 1).

In bagging, the samples in each base classifier training set are chosen with replacement, so elements may be repeated in the training "set" for a single base classifier. Bagging has been proposed as an *empirical* defense against poisoning attacks (Biggio et al., 2011) as well as for evasion attacks (Smutz & Stavrou, 2016). However, to our knowledge, these techniques have not yet been used to provide *certified* robustness.

Our approach also bears some similarity to Federated Averaging (McMahan et al., 2017) in that base models are trained on disjoint partitions of the dataset. However, in Federated Averaging, the *model weights* of many distributed base classifiers are periodically averaged and re-distributed during learning, in order to allow for efficient massively-parallel learning. No theoretical robustness guarantees are provided (as this is not the goal of the algorithm) and it would seem difficult to derive them, given that the relationship between model weights and final classification is highly non-linear in deep networks. By contrast, DPA uses the consensus of *final outputs* after all base classifiers are trained independently (or, in the SS-DPA case, independently for labeled data).

| | Number of Partitions $k$ | Median Certified Robustness | | Clean Accuracy | | Base Classifier Accuracy | |
|---|---|---|---|---|---|---|---|
| | | **Hash** | **Sort** | **Hash** | **Sort** | **Hash** | **Sort** |
| **MNIST, SS-DPA** | 1200 | 460 | **485** | **95.74%** | 95.62% | 78.64% | **80.77%** |
| | 3000 | 606 | **645** | 93.87% | **93.90%** | 55.04% | **57.65%** |
| **CIFAR, SS-DPA** | 50 | **25** | 25 | **90.90%** | 90.89% | 89.00% | **89.06%** |
| | 250 | **124** | 124 | **90.36%** | 90.33% | **86.34%** | 86.25% |
| | 1000 | 387 | **392** | **89.11%** | 89.02% | 75.35% | **75.83%** |
| **GTSRB, SS-DPA** | 50 | **25** | 25 | 96.98% | **97.09%** | 96.33% | **96.35%** |
| | 100 | **50** | 50 | **96.79%** | 96.76% | 94.86% | **94.96%** |
| | 200 | **99** | 99 | **96.38%** | 96.34% | 91.39% | **91.54%** |
| | 400 | 174 | **176** | **95.89%** | 95.80% | 83.07% | **83.60%** |

Table 2: Comparison of SS-DPA with hashing ('**Hash**' columns, described in Appendix E) to the SS-DPA algorithm with partitions determined by sorting ('**Sort**' columns, described in the main text). Note that partitioning via sorting consistently results in higher base classifier accuracies, and can increase (and never decreases) median certified robustness. These effects seem to be larger on MNIST than on CIFAR-10.

## E   SS-DPA with Hashing

It is possible to use hashing, as in DPA, in order to partition data for SS-DPA: as long as the hash function $h(\boldsymbol{t})$ does not use the sample label in assigning a class (as ours indeed does not), it will always assign an image to the same partition regardless of label-flipping, so only one partition will be affected by a label-flip. Therefore, the SS-DPA label-flipping certificate should still be correct. However, as explained in the main text, treating the unlabeled data as trustworthy allows us to partition the samples evenly among partitions using sorting. This is motivated by the classical understanding in machine learning (e.g. Amari et al. (1992)) that learning curves (the test error versus the number of samples that a classifier is trained on) tend to be convex-like. The test error of a base classifier is then approximately a convex function of that base classifier's partition size. Therefore, if the partition size is a random variable, by Jensen's inequality, the expected test error of the (random) partition size is greater than the test error of the mean partition size. Setting all base classifiers to use the mean number of samples should then maximize the average base classifier accuracy. To validate this reasoning, we tested SS-DPA with partitions determined by hashing (using the same partitions as we used in DPA), rather than the sorting method described in the main text. See Table 2 for results. As expected, the average base classifier accuracy decreased in most (8/9) experiments when using the DPA hashing, compared to using the sorting method of SS-DPA. However, the effect was minimal in CIFAR-10 and GTSRB experiments: the main advantage of the sorting method was seen on MNIST. This is partly because we used more partitions, and hence fewer average samples per partition, in the MNIST experiments: fewer average samples per partition creates a greater variation in the number of samples per partition in the hashing method. However, CIFAR-10 with $k = 1000$ and MNIST with $k = 1200$ both average 50 samples per partition, but the base classifier accuracy difference still was much more significant on MNIST (2.13%) compared to CIFAR-10 (0.48%).

On the MNIST experiments, where the base classifier accuracy gap was observed, we also saw that the effect of hashing on the smoothed classifier was mainly to decrease the certified robustness, and that there was not a significant effect on the clean smoothed classifier accuracy. As discussed in Appendix F, this may imply that the outputs of the base classifiers using the sorting method are more correlated, in addition to being more accurate.

## F   Effect of Random Seed Selection

In Section 3.2.1, we mention that we "deterministically" choose different random seeds for training each partition, rather than training every partition with the same random seed. To see a comparison between using distinct and the same random seed for each partition, see Table 3. Note that there is not a large, consistent effect across experiments on either the base classifier accuracy nor the the median certified robustness: however, in most experiments, the distinct random seeds resulted in

| | Number of Partitions $k$ | Median Certified Robustness | | Clean Accuracy | | Base Classifier Accuracy | |
|---|---|---|---|---|---|---|---|
| | | **Same** | **Distinct** | **Same** | **Distinct** | **Same** | **Distinct** |
| **MNIST, DPA** | 1200 | 443 | **448** | 95.33% | **95.85%** | 76.21% | **76.97%** |
| | 3000 | **513** | 509 | 92.90% | **93.36%** | 49.52% | **49.54%** |
| **MNIST, SS-DPA** | 1200 | **501** | 485 | 94.45% | **95.62%** | **82.31%** | 80.77% |
| | 3000 | **663** | 645 | 91.60% | **93.90%** | **59.33%** | 57.65% |
| **CIFAR, DPA** | 50 | **9** | **9** | 70.14% | **70.16%** | 56.13% | **56.39%** |
| | 250 | **5** | **5** | 55.24% | **55.65%** | 34.96% | **35.17%** |
| | 1000 | N/A | N/A | 43.96% | **44.52%** | 23.14% | **23.20%** |
| **CIFAR, SS-DPA** | 50 | **25** | **25** | **90.92%** | 90.89% | **89.08%** | 89.06% |
| | 250 | **124** | **124** | 90.25% | **90.33%** | **86.25%** | **86.25%** |
| | 1000 | 391 | **392** | 88.97% | **89.02%** | 75.82% | **75.83%** |
| **GTSRB, DPA** | 50 | **20** | **20** | 88.59% | **89.20%** | 73.89% | **73.94%** |
| | 100 | 3 | **4** | 54.98% | **55.90%** | 34.74% | **35.64%** |
| **GTSRB, SS-DPA** | 50 | **25** | **25** | 97.06% | **97.09%** | **96.36%** | 96.35% |
| | 100 | **50** | **50** | **96.79%** | 96.76% | 94.95% | **94.96%** |
| | 200 | **99** | **99** | **96.34%** | **96.34%** | **91.54%** | **91.54%** |
| | 400 | **176** | **176** | **95.85%** | 95.80% | **83.64%** | 83.60% |

Table 3: Comparison of DPA and SS-DPA algorithms using the same random seed for each partition ('**Same**' columns, described in Appendix F) to the DPA and SS-DPA algorithms using the distinct random seeds for each partition ('**Distinct**' columns, described in the main text). Note that using distinct random seeds usually results in higher smoothed classifier clean accuracies, and always does so when using DPA.

higher smoothed classifier accuracies. This effect was particularly pronounced using SS-DPA on MNIST: using distinct seeds increased smoothed accuracy by at least 1% on each value of $k$ on MNIST. This implies that shared random seeds make the base classifiers *more correlated* with each other: at the same level of average base classifier accuracy, it is more likely that a plurality of base classifiers will all misclassify the same sample (If the base classifiers were perfectly uncorrelated, we would see nearly 100% smoothed clean accuracy wherever the base classifier accuracy was over 50%. Also if they were perfectly correlated, the smoothed clean accuracy would equal the base classifier accuracy). Interestingly, the base classifier accuracy was significantly lower for both SS-DPA/MNIST experiments when using diverse random seeds; this defies any obvious explanation. However, this does make the correlation effect even more significant: for example, for $k = 3000$, the SS-DPA smoothed classifier accuracy is over 2% larger with distinct seeds, despite the fact that the base classifier is over 1% less accurate.

It is somewhat surprising that the base classifiers become correlated when using the same random seed, given that they are trained on entirely distinct data. However, two factors may be at play here. First, note that the random seed used in training controls the random cropping of training images: it is possible that, because the training sets of the base classifiers are so small, using the same cropping patterns in every classifier would create a systematic bias.

Note that the effect was least significant for SS-DPA on CIFAR-10 and GTSRB, with SimCLR. This may be due to the final supervised classifiers being linear classifiers, rather than deep networks, for these experiments.

## G  SS-DPA WITH REPEATED UNLABELED DATA

The definition of a training set that we use, $T \in \mathcal{P}(\mathcal{S}_L)$, technically allows for repeated samples with differing labels: there could be a pair of distinct samples, $t, t' \in T$, such that $\textbf{sample}(t) = \textbf{sample}(t')$, but $\textbf{label}(t) \neq \textbf{label}(t')$. This creates difficulties with the definition of the label-flipping attack: for example, the attacker could flip the label of $t$ to become the label of $t'$: this would break the definition of $T$ as a set. In most applications, this is not a circumstance that warrants practical consideration: indeed, none of the datasets used in our experiments have such instances (nor can

| | Number of Partitions $k$ | | Median Certified Robustness | | Clean Accuracy | | Base Classifier Accuracy | |
|---|---|---|---|---|---|---|---|---|
| | | | **Equal.** | - | **Equal.** | - | **Equal.** | - |
| **GTSRB, DPA** | 50 | | **23** | 20 | **91.84%** | 89.20% | **80.98%** | 73.94% |
| | 100 | | **21** | 4 | **77.81%** | 55.90% | **54.83%** | 35.64% |

Table 4: Comparison of DPA algorithm with and without using histogram equalization as preprocessing on the GTSRB dataset. We find that histogram equalization substantially improves performance when $k = 100$, although the effect is more modest at $k = 50$.

label-flipping attacks create them), and therefore, for performance reasons, our implementation of SS-DPA does not handle these cases. Specifically, to optimize for performance, we verify that there are no repeated sample values, and then sort $T$ itself (rather than $\mathbf{samples}(T)$) lexicographically by image pixel values: this is equivalent to sorting $\mathbf{samples}(T)$ if no repeated images occur — which we have already verified — and avoids an unnecessary lookup procedure to find the sorted index of the unlabeled sample for each labeled sample. However, the SS-DPA algorithm as described in Section 3.3 can be implemented to handle such datasets, under a formalism of label-flipping tailored to represent this edge case.

Specifically, we define the space of possible labeled data points as $\mathcal{S}'_L = \mathcal{S} \times \mathcal{P}(\mathbb{N})$: each labeled data point consists of a sample along with a *set* of associated labels. We then restrict our dataset $T$ to be any subset of $\mathcal{S}'_L$ such that for all $\boldsymbol{t}, \boldsymbol{t}' \in T$, $\mathbf{sample}(\boldsymbol{t}) \neq \mathbf{sample}(\boldsymbol{t}')$. In other words, we do not allow repeated sample values in $T$ as formally defined: if repeated samples exist, one can simply merge their sets of associated labels (in the formalism).

Note that using this definition, the size of $T$ will equal the size of $\mathbf{samples}(T)$, and the samples will always remain the same: the adversary can only modify the label sets of samples "in place". SS-DPA will always assign an unlabeled sample, *along with all of its associated labels*, to the same partition, regardless of any label flipping. In practice, this is because, as described in Section 3.3, the partition assignment of a labeled sample depends only on its sample value, not its label: all labeled samples with the same sample value will be put in the same partition. Note that this is true even if the implementation represents two identical sample values with different labels as two separate samples: one does not actually have to implement labels as sets. Therefore, any changes to any labels associated with a sample will only change the output of one base classifier, so all such changes can together be considered a single label-flip in the context of the certificate. In the above formalism, the certificate represents the number of samples in $T$ whose label *sets* have been "flipped": i.e., modified in any way.

## H    GTSRB with Histogram Equalization

Because GTSRB contains images with widely varying lighting conditions, histogram equalization is sometimes applied as a preprocessing step when training classifiers on this dataset (for example, Two Six Labs (2019)). We tested this preprocessing with DPA, and found that it substantially improved the performance for larger $k$ ($k = 100$). However, for $k = 50$, which had larger accuracy and median certified robustness with or without this preprocessing, the effect was modest (a 2.64% increase in clean accuracy, and an increase of 3 in certified robustness). The greater effect when $k$ is large is likely because each partition contains fewer images of each class, so it is more likely for each base classifier that some lighting conditions are not represented in the training data for every class. Applying histogram equalization to the training and test data reduces this effect.

## I    SimCLR Experimental Details

We used the PyTorch implementation of SimCLR provided by Khosla et al. (2020), with modifications to ensure determinism as described in the main text. For training the embeddings, we used a ResNet18 model with batch size of 512 for CIFAR-10 and 256 for GTSRB, initial learning rate of 0.5, cosine annealing, and temperature parameter of 0.5, and trained for 1000 epochs. For learning

the linear ensemble classifiers, we used a batch size of 512, initial learning rate of 1.0, and trained for 100 epochs.

## J    GTSRB DATASET DETAILS

The GTSRB dataset consists of images of variable sizes, from $15 \times 15$ to $250 \times 250$. We resized all images to $48 \times 48$ during training and testing (using bilinear interpolation). However, we wanted to ensure that our certificates still applied correctly to the original images. In particular, for SS-DPA, we sort the dataset using pixel values of the original image, with black padding for smaller images. This ensures that repeated images do not occur in the original dataset, (as described in Appendix G), even if the resized images could possibly contain repeats. For DPA, we hash using the sum of all pixels in the original image. Finally, for sorting to ensure determinism in training, we use the images after resizing (ensuring no repeated images is not important for this). For both NiN and SimCLR training, we excluded horizontal flips from data augmentation and contrastive learning, because some classes in GTSRB are in fact mirror-images of other classes.

