# OpenReview forum: "Deep Partition Aggregation: Provable Defenses against General Poisoning Attacks"
_ICLR.cc/2021/Conference — ICLR 2021 Poster_

### Official Review · AnonReviewer1 · 2020-10-28
**Review for Deep Partition Aggregation: Provable Defenses against General Poisoning Attacks**

**Rating:** 7
**Confidence:** 4

**Review:**

This paper studies how to enhance the robustness of classifiers in face of data poisoning attacks. The key insight of the paper is that adding or deleting one training point can at most change one of the k partitions of the training set. Based on this idea, the authors propose Deep Partition Aggregation (DPA), a robust classification algorithm that first partitions the training data into k subsets, and then separately train a model on each subset. The final prediction is an aggregation of the predictions of those k classifiers using the majority vote. Apart from DPA, the authors also consider a setting where a large amount of data points do not have labels. In that scenario, semi-supervised learning algorithms are used as based algorithm when training separate models on each subset. The proposed method SS-DPA enjoys the property of being fast to train, since learning is performed on a subset that has smaller number of data points. The paper derives theoretical guarantees in terms of when the prediction of a particular data point can be certifiably correct. Finally, experiments on MNIST and CIFAR demonstrate the effectiveness of the proposed defense as compared to prior works.

The main advantage of the paper lies in the experimental part. Empirically, the proposed method indeed shows strong defense ability against poisoning attacks. For example, on the MNIST dataset, more than half of the test images can be certified with DPA. Furthermore, by comparison with prior works, the authors show that the proposed DPA and SS-DPA are better at defending against data poisoning attacks.

One disadvantage of the paper is that the power of the attacker in this paper seems to be very weak. For example, for MNIST, the attacker only changes the label of 1% training data. However, even with such a small fraction of change, the proposed DPA already suffers 0 certified accuracy. This is a bit disappointing because 1% does not seem to be a large fraction. I am wondering if the authors could justify that 1% poisoning in practice is already a significant amount of change to the training data, thus is unlikely to happen in the real world. That will convince me that the proposed defense is indeed useful for a real-world scenario.

Apart from the basic DPA, the paper also proposed another algorithm SS-DPA that applies when there is a large number of unlabeled data points, i.e., a semi-supervised learning setting. I am wondering to what extend unlabeled helps with improving the certified robustness? From a theoretical perspective, the classification is only about the discriminative part of the data distribution, and there is theory in semi-supervised learning that suggests using unlabeled data points do not substantially improve the accuracy, e.g., the following paper

Does Unlabeled Data Provably Help?
Worst-case Analysis of the Sample Complexity of Semi-Supervised Learning

Thus I am wondering if the authors could explain why unlabeled data points should be used in the defense problem?
Theorem 1 and Theorem 2 look pretty much the same. I am wondering if the difference only lies in the baseline classifier, one being supervised and the other being semi-supervised? If so, I think the authors should consider merging the two theorems, as they are not quite different from each other.

Finally, it is not clear how to pick the parameter k, i.e., the number of partitions. The experimental results seem to suggest that the larger k is, the better robustness guarantee we will have. However, k cannot go to infinitely large. One has to at least ensure that each base classifier has decent accuracy, which requires a reasonably large number of training data in each partition. Therefore, k cannot be too large. I am wondering if the authors could provide some empirical guidance regarding how to pick k?

---

> ### Author Response · Authors · 2020-11-21
> **Rebuttal to Reviewer #1**
>
> We thank you for your comments:
>
> -- “For example, for MNIST, the attacker only changes the label of 1% training data. However, even with such a small fraction of change, the proposed DPA already suffers 0 certified accuracy”: This is inaccurate: on MNIST, we are able to certify over 50% of images to 600 (= 1%) label changes. For general poisoning, we can still certify over 30% of test samples (not zero) to 600 insertions or removals. Additionally, we must consider that 600 changes represents 10% of the source (or target) class, so is not an insignificant change. We note that these results establish new SOTA in this dataset.
>
> -- “theory in semi-supervised learning that suggests using unlabeled data points do not substantially improve the accuracy”: Despite theoretical results showing that unlabeled data may not help in the worst case, the fact remains that semi-supervised learning is still a widely used technique, which empirically improves accuracy. There is a wide range of literature demonstrating this, which we cite.
>
> --- “Theorem 1 and Theorem 2 look pretty much the same.”: there are two important differences. First, in Theorem 2, we use sorting, rather than hashing, to ensure that each partition gets a similar number of samples: this improves performance (see Appendix E) and is only possible in the label-flipping case. Second, in the label-flipping case we can ensure that the label-flip does not re-assign the partition that a sample falls in (simply by not using the label in assignment) so the guarantee is in terms of the number of label flips, rather than the symmetric difference (which would be double)
>
> --- The hyperparameter k: the reviewer is correct that the hyperparameter k controls a tradeoff between robustness and accuracy. In our experiments (including new GTSRB experiments which we have added in Figure 4) we show performance over a range of values of k.

---

### Official Review · AnonReviewer3 · 2020-10-28
**A neat idea to merge semi-supervised learning and ensemble learning in the context of Adversarial Poisoning. Might need some more substantiation.**

**Rating:** 6
**Confidence:** 3

**Review:**

The paper proposes to solve two variants of adversarial poisoning attacks: 1) General Poisoning Attacks - where either the input is distorted or the label is flipped. 2) Label Flipping Poisoning Attacks - where the input images are intact but only the labels are flipped. The crux of the algorithm is the standard ensemble model idea, where we train multiple models each with a partition of the data and take the most prevalent prediction among all models during inference. This naturally adds robustness to the Additionally, the paper also provides theoretical lower bounds on the 'amount' of distortion below which precision is preserved.

For the case of general poisoning attacks, the paper proposes a certified defense against the cardinality of the symmetric distance of the original training dataset and the poisoned dataset (including both image distortion and/or label flips). For the special case of label-flip attacks only, the certification follows Rosenfeld et al. 2020 where we quantify the minimum number of label flips required to change the prediction of a particular sample point.

The algorithm for the general poisoning case (called DPA - Deep Partition Aggregation) is simple. Just assign each training sample to one of K partitions based on a random hash function. Then train K classifiers on their respective data subsets. During inference, we just need to get a label prediction from each of the K models and get the mode of all those predictions.

When the attacks are restricted to just label flips, we can simply invoke semi-supervised learning. For each of the K models, we can first train an unsupervised model on the entire training data (since inputs are not corrupted). This will map inputs to a nice semantic space without any label information. Then the partitioned data labels can be used to train individual models like before. Any label flip can only effect one of the K models. Hence, the inference is robust as it is an aggregation over several models.

The results on MNIST and CIFAR are very encouraging and SOTA as per the papers claims.

Pros:
1. strong experimental results
2. paper clearly written with nice introduction to types of attacks.
3. Theoretical bounds on tolerance to adversarial attacks.

Cons:
1. Similarities with federated learning methods - I've some novelty concerns. Federated Averaging (McMahan et al. 2017) hinges on a very similar idea of partitioning labels (not disjointly but with some overlap). SO the novelty here seems to be only the application?
2. How abt evasion attacks? - There is no discussion about how this method tackles evasion attacks or what prevents it from tackling it.
3. repetitive plots- Figure 1 plots are reappearing in Figure 2. It is good for a reader but might make the paper look less substantial.
4. fairly simple idea - notwithstanding the federated averaging idea, ensemble methods are common in several walks of ML. For example,  (Medini et. al. 2019, NeurIPS), uses random hash functions to create label partitions to train with 50 MM labels. even the concept of using the model number as a random seed to avoid correlations seems to be prevalent in that paper. Infact, feature hashing and JL-Lemma literature use it in and out.
5. might warrant experiments on more datasets

---

> ### Author Response · Authors · 2020-11-21
> **Rebuttal to Reviewer #3**
>
> We thank you for your comments:
>
> --- Similarity to Federated Averaging:  In Federated Averaging, the _model weights_ of many distributed learners are periodically averaged and re-distributed during learning, in order to allow for efficient massively-parallel learning. No theoretical robustness guarantees are provided (as this is not the goal of the algorithm) and it would seem difficult to derive them, given that the relationship between model weights and final classification is highly non-linear in deep networks. By contrast, we use the consensus of _final outputs_ after all models are trained independently (or, in the SS-DPA case, independently for labeled data): this does provide robustness guarantees. We have added a reference to FA as an example of another ensemble technique that partitions data into Appendix D.
>
> --- Evasion attacks: We do not claim to provably defend against evasion attacks. Such guarantees likely do not exist: it is possible (although highly unlikely) for example that all trained models in our ensemble are identical, in which case the problem reduces to evading a single arbitrary model. However, as mentioned in the paper, in practice, ensemble methods have been used to empirically defend against evasion attacks  (Smutz and Stavrou, 2016)
>
> -- Redundant plots: We have rearranged plots to fix this issue, using Figure 1 only for comparison to prior work.
>
> --- Similarity to Medini et al: Medini et al uses random hashing on _classes_ to create a hierarchy of classifiers, each of which classifies into arbitrary buckets of combined classes, in order to deal with very large numbers of possible classes: each classifier is trained on all training data. By contrast, our method uses hashing on individual training samples to determine which classifier to train with the training sample.
> We do not find any mention of random seed selection for training deep networks in Medini et al.
>
> -- Additional datasets: We have added results on GTSRB,  where there are 43 classes with fewer than 1000 samples per class on average: here, we can certify against 20 general poisonings (in median), or 177 label flips.
>
> |Method |Number of Partitions k | Median Certified Robustness | Clean Accuracy |
> |--|--|--|--|
> |DPA| 50|20|88.43%|
> |DPA|100|3|55.69%|
> |--|--|--|--|
> |SS-DPA|50|25|97.39%|
> |SS-DPA|100|50|97.22%|
> |SS-DPA|200|99|96.87%|
> |SS-DPA|400|177|96.18%|

---

### Official Review · AnonReviewer4 · 2020-10-31
**Strong work on certified robustness (for individual test instances) against data poisoning**

**Rating:** 8
**Confidence:** 4

**Review:**

Context and summary of the results:

Data poisoning attacks deal with adversaries who change the training set, up to a certain degree (controlled in different ways), with the aim of lowering the "quality" of the produced model. Previously methods were designed to tolerate adversarial perturbations while preserving low risk in the produced model.

This paper studies point-wise certification methods for the decision made on a test instances against data poisoning.  Namely, for a given point, we would like to know: how many changes in the training set would still lead to the same produced label. Knowing this result gives confidence about the predicted label, even if some poisoning has already happened (to a certain degree). Exhaustive search (on various cases of data and retraining) would not be possible, and a theoretically sound algorithmic method would be necessary.

This paper proposes a general, simple, yet effective idea for certification in poisoning context: partition the input data into a bunch of subsets and then train sub-models on them independently; then output the majority of the predictions. It is not hard to see that if the majority vote is far by amount alpha from the 2nd majority vote, then one can certify up to alpha/2 changes in the training set. To make this proposal formal, there are some subtleties that the paper handles by using hash functions to do the partition (so that the order of the elements won’t affect which subset they land). This approach is indeed reminiscent of "Bagging" (Bootstrap Aggregating), but this seems to be the first work (along with concurrent cited works) that apply this to certification under poisoning attacks.

Other than studying “general” poisoning attacks (in which changes in the data set are measured by Hamming distance), the paper further applies the idea to the special case of label-flip attacks, which were previously studied by Rosenstein et al. as well, but this work shows how to improve their bound through the experiments on popular data sets quite strongly. Along the way, (for label flipping) there are some neat ideas also to improve the results and the efficiency. (1) The paper shows how to benefit from semi-supervised learning by each of the sub-models is still trained on the *whole* data set, while only a subset of the samples (i.e., those in a particular partition) would have the labels kept. (2) To make this scalable, the paper carefully picks the semi-supervised method in such a way that all trainings of sub-models share the same 1st step of dealing with non-labeled data (on the *whole* data set).

To evaluate: Certification of decisions under poisoning is a very recent and important direction. The results of this work are convincing as they improve the bounds achieved in previous work quite strongly. The methods used here, though seemingly simple in the hindsight, are sound and might be applicable in other settings as well. The paper is written quite clearly. I, therefore, recommend acceptance.

Comments:

The paper mentions that
"Rosenfeld et al. (2020) does not provide a provable defense for this more general case."
However, it seems the latest version of this previous work of Rosenstein et al. (from Aug'20 - 2nd paragraph of section 4) actually did propose a method for "general" poisoning attacks using randomized smoothing as well, even though they only applied the idea concretely only to label flipping. So, even though it is not quite clear how their exact certification bounds would be, and also the result would be randomized certification as opposed to deterministic, which is the result of this paper, I still think this previous work on general poisoning certification shall be mentioned as well.

Some related works: The exact noise models studied in this paper are actually studied long ago in classical learning papers that, unfortunately, do not get to be cited in recent works in poisoning attacks much. It is true that those works are not about certification, but the basic problem of provable learning under poisoning/noise are essentially the same. Citation [1] below defined the label flip noise model, and [2] defined the "general" poisoning model in which the adversary can change a fraction of the data. The current paper actually uses a more fine-grained version that distinguishes between add/removes, but this is the same as measuring Hamming distance up to a factor 2. Works [3,4] also initiated a computationally efficient approach to dealing with poisoning attacks in a provable way, and [6] particularly studied supervised learning and poisoning-robust SGC. Of course, I understand that here we want an extra certification on top of the defense, yet I still think the line of work on provable defenses against poisoning are closely related. Also, I think your work (and similar certified robustness papers on general poisoning) can be interpreted as "certified poisoning against *targeted* poisoning" because you certify the decisions for each individual test instance. So, they seem to complement "provable attacks" on targeted poisoning e.g., from [5] in which it is shown how to increase the error close to one for a particular test instance if the original error (without attack) is a not-so-small probability to begin with. It seems one difference between the two settings is that the attack of [5] needs a bigger poisoning budget to succeed and you certify a smaller number (naturally) but I think a comparison with the complementary line of work is needed as well.

I also think that the comparison with Steinhardt et al. is better to be expanded a bit more. Both papers use the term "certified robustness" but as (correctly) mentioned in the paper, the two settings are not the same (e.g., here the certification is for a test instance rather than the model). I think it is better to bring this note out of a footnote and shed more light on this helpful comparison to prevent confusion for readers.

I know that the experiments are done with image datasets, but the problem studied in this work is more general, and it would have been probably better if the treatment of the issue was also as such (e.g., the abstract mentions "bounded number of images", though it could be basically any piece of data).


Citations:

[1] Sloan, Robert H. "Four types of noise in data for PAC learning." Information Processing Letters, 1995.

[2] Bshouty, Nader H., Nadav Eiron, and Eyal Kushilevitz. "PAC learning with nasty noise." Theoretical Computer Science 2002.

[3] Ilias Diakonikolas, Gautam Kamath, Daniel M Kane, Jerry Li, Ankur Moitra, and Alistair Stewart. "Robust estimators in high dimensions without the computational intractability." FOCS, 2016

[4] Kevin A Lai, Anup B Rao, and Santosh Vempala. "Agnostic estimation of mean and covariance." FOCS 2016.

[5] Mahloujifar Saeed, Dimitrios I. Diochnos, and Mohammad Mahmoody. "The curse of concentration in robust learning: Evasion and poisoning attacks from concentration of measure." AAAI 2019.

[6] Diakonikolas, I., Kamath, G., Kane, D., Li, J., Steinhardt, J., & Stewart, A. "Sever: A robust meta-algorithm for stochastic optimization." ICML. 2019.

*********** post rebuttal comment ************

Thanks for the comments and clarifiactions.
Just to add that: I agree that Rosenfeld et al (2020)'s work on general poisoning is concurrent to yours (as that part is expanded in sub-sequent updates to their Arxiv, not the original ICML paper).  And anyway the bounds of this submission are stronger. Yet, I think that a brief discussion/comparison with this aspect of the  Rosenfeld et al (2020) paper is helpful to the reader.

---

> ### Author Response · Authors · 2020-11-21
> **Rebuttal to Reviewer #4**
>
>
> We thank you for your very detailed comments.
>
> -- We have included a mention to the “general poisoning” defense proposed by Rosenfeld et al (2020): this is concurrent and independent to our work. The “general poisoning” discussed in that work is not specifically defined, however; for example, the $L_2$ defense proposed by Weber et al (2020) falls within the framework discussed. Also, the “general poisoning” defense proposed by Rosenfeld et al (2020) is not discussed in a de-randomized context, while our method is de-randomized.
>
> --- We have expanded our related works section to discuss references [1-6].
>
> --- We have fixed the language “images” to “training samples” where appropriate.

---

### Official Review · AnonReviewer2 · 2020-11-03
**Simple defense strategy that suffers from poor empirical performance and large drops in clean accuracy**

**Rating:** 4
**Confidence:** 4

**Review:**

**Pros:**
+ The paper is clearly written and easy to follow.
+ Provides some certifiable measure of robustness against general poisoning attacks

**Cons:**
- The extremely simplistic defense strategy that is utilized implies that the defense has very poor performance. On the CIFAR-10 dataset, the drop in clean accuracy is >20% but only around 9 samples out of 50,000 can be certified robust. These results consider a very weak attacker poisoning less than 50 samples out of 50,000. Thus, while defense may claim to provide some certified robustness against general poisoning attacks, the results are not promising. There is also no discussion of how the defense can be improved.
- The proof of theorem 1 appears to be a missing a critical piece, which is the determination of how the quantity $\rho(x)$ was actually computed.

---

> ### Author Response · Authors · 2020-11-21
> **Rebuttal to Reviewer #2**
>
> We thank you for your comments. While our performance for general poisoning attacks on CIFAR-10 is admittedly not very impressive, we emphasize that this work is the “first” work to certify for attacks in this threat model *at all*. We also note that we outperform the previous SOTA for “label flip” attacks on CIFAR-10 by a large margin (392 label flips, compared to 175). Finally, even in the general case, we are able to defend, in median, against a very large number of attacks (509) on MNIST.
>
> We have also added results for the GTSRB dataset in Section 4, where there are 43 classes with fewer than 1000 samples per class on average: here, we can certify against 20 general poisonings (in median), or 177 label flips.
>
> |Method |Number of Partitions k | Median Certified Robustness | Clean Accuracy |
> |--|--|--|--|
> |DPA| 50|20|88.43%|
> |DPA|100|3|55.69%|
> |--|--|--|--|
> |SS-DPA|50|25|97.39%|
> |SS-DPA|100|50|97.22%|
> |SS-DPA|200|99|96.87%|
> |SS-DPA|400|177|96.18%|
>
> The formula for $\rho(x)$ is given in Equation 5. It is derived by solving the inequality in the second- to-last line of the proof of Theorem 1 (on page 13).

---

### Author Response · Authors · 2021-03-18
**Camera-Ready Revision**

Due to minor changes in the code to more accurately ensure deterministic execution (in particular, corrections to the use of the cudnn.benchmark flag), some reported numeric results have changed slightly in the camera-ready: these minor changes do not effect our conclusions.

---

### Decision · Program_Chairs · 2021-01-07
**Final Decision**

**Decision:**

Accept (Poster)

**Comment:**

The authors develop a novel strategy, Deep Partition Aggregation, to train models to be certifiably robust to data poisoning attacks based on flipping labels of a small subset of the training data or introducing poisoned input features. They improve upon existing certified defences against data poisoning and are the first to establish certified guarantees against general poisoning attacks.

Most reviewers were in support of acceptance. Reviewer concerns were raised in the rebuttal phase but were convincingly addressed in the rebuttal phase. One reviewer did raise concerns on the weakness of experimental results on CIFAR-10, but the fact that this method has established the first certified defence in the general poisoning setting and that the results are stronger on other datasets certainly warrant acceptance. I would encourage the authors to clarify this in the final version.